# Are Job Demands Necessary in the Influence of a Transformational Leader? The Moderating Effect of Role Conflict

**DOI:** 10.3390/ijerph18073630

**Published:** 2021-03-31

**Authors:** Pedro A. Díaz-Fúnez, Carmen M. Salvador-Ferrer, Natalia García-Tortosa, Miguel A. Mañas-Rodríguez

**Affiliations:** IPTORA Research Team, University of Almeria, 04120 Almeria, Spain; cmsalva@ual.es (C.M.S.-F.); ngt775@gmail.com (N.G.-T.); marodrig@ual.es (M.A.M.-R.)

**Keywords:** intellectual stimulation, role conflict, intellectual engagement, performance

## Abstract

(1) Background: The objective of this manuscript is to propose the necessity of job demands to ensure the positive influence of policies in stimulating employees’ engagement and performance. If the policies related to the intellectual stimulation of employees implemented by team leaders are to have positive effects on employee performance, they must induce emotional engagement in the employees. Furthermore, to achieve this positive influence on emotions, the organization must offer an environment that challenges the employees in the organization. Here, we analyze a moderate mediation model to examine the moderating, positive effect of role conflict on the intellectual engagement and performance of employees. (2) Methods: This study involved 705 employees of a multinational private company based in Spain. (3) Results: We confirm the positive moderating effect of role conflict between the intellectual stimulation of employees and intellectual engagement, and the mediating effect of intellectual engagement between leadership behavior and employee performance. (4) Conclusions: Organizational leader stimulation practices necessitate an environment of moderate job demands in order to improve the intellectual engagement of employees, thereby increasing their performance. The implications of the findings in terms of theory, research and practice are discussed.

## 1. Introduction

In recent years, research on transformational leadership has suggested the crucial role of leaders within the organizational context. Leader behavior influences the way people experience their jobs [1]. On the one hand, leaders need to develop and motivate their workers to ensure each of them is willing to take care of their own tasks; on the other hand, they have to make tasks interesting and attractive enough not to fall into routine and boredom [2]. However, is the leader’s behavior alone enough to incentivize employee performance? It seems not; as Shields [3] states, good intentions are not enough. In the organizational context there are other factors that can facilitate or hinder the work of transformational leadership policies.

A theory that addresses how the factors of organizational context affect leadership actions is the job demands–resources theory (JD-R) [4]. This proposed theory has positive relevance in the organizational context, allowing us to analyze how negative aspects of the context determine the influence of the leader’s behaviors on their collaborators. The negative elements of an organization are referred to as job demands, and are defined as aspects of work that require effort and are associated with physical and psychological costs for the worker. Traditionally, a role’s stress dimensions have been studied in terms of the main demands of work in relation to leadership behavior [5]. A role is “a set of expectations about behaviour for a position” in an organization [6] (p. 155), and role stressors are the socio-psychological or behavioral restrictions ascribed to a role [7]. Role conflict is a core dimension of role stress, and it is defined as incongruences or incompatibilities in the requirements of the role [6]. In multinational companies, the variety of tasks requires that employees evaluate the time requirements, making sure they do not exceed their abilities to cope with them, which would mean they suffer from increased tension, resulting in stress responses. However, the JD-R theory offers increasing evidence that job demands do not only generate negative results; in some situations, they can cause employees to elevate their skills to meet those demands, therefore outlining the influence of leadership on collaborators [8].

In this sense, role conflict can be necessary when it comes to achieving the balance between employee development, leadership, and the difficulty of a task. This type of job is known as “active jobs” [9]. Active work contexts are characterized by high job demands, but in which employees are provided with high levels of control. These contexts are predictors of positive results for employees, such as increased work engagement or motivation through skills development [10]. When developing active work contexts, labor demands, such as role conflict, can play a fundamental role in the influence of leadership over work engagement improvement and performance.

Therefore, the objective of this study is to analyze the conditions under which the transformational leadership dimension of intellectual stimulation influences performance through intellectual engagement, while examining how the relationship between the intellectual stimulation of the leader and intellectual engagement is affected by the modulating effect of role conflict. In other words, employees will feel intellectually engaged when their team leader induces intellectual stimulation and when they experience role conflict in the organizational context. All this will positively affect the performance of employees.

According to Burns [11], transformational leadership is a dynamic and bidirectional process in which both leaders and followers transform each other. This type of leadership “transforms” the norms and values of employees, and motivates them to perform beyond their own expectations [12]. Intellectual stimulation is one dimension of transformational leadership [13], focusing on leadership behaviors that motivate followers’ intellectual curiosity, encourage the re-examination and challenging of old assumptions, and reward new thinking and innovative approaches [14]. When followers feel intellectually stimulated, they approach problems in new ways and make more creative solutions [15]. This mindset allows followers to contribute novel, useful ideas, as they recognize the benefits and/or detriments of their own and others’ perspectives [16]. Followers who experience intellectual stimulation also become more confident in contributing ideas that are complex or unusual [17], manifesting new ways of dealing with demands.

When it comes to analyzing the way in which leadership affects organizational results, studies refer to the mediating effect of engagement. Work engagement is a positive affective–motivational state related to work in which employees are willing to make extra efforts, feel good, and are able to separate themselves from their work and maintain a healthy balance between work and private life [18]. There are many educational studies that have linked intellectual stimulation with intellectual engagement in learning processes [19,20], but few studies have considered whether this influence also manifests in the organizational context [21]. Intellectual engagement is defined as “the extent to which one is intellectually absorbed in work and thinks about ways to improve work” [22] (p. 8). This variable involves the initiation and focused application of cognitive effort in obtaining a goal or the solution to a challenge.

In the organizational context, the intellectual engagement of those who are managed exclusively via transactional leadership may not facilitate the expected levels of creativity and performance [23]. Bass et al. [24] claim that the transformational leader who is intellectually stimulating will encourage creativity and problem-solving. For instance, a manager who rejects transformational leadership and practices transactional leadership only may restrict the freedom of their subordinates, and thus hamper their creativity and innovation skills [25]. In the context of knowledge management, Jansen et al. [26] argue that creativity, taking greater risks and developing new ideas are more commonly related to the intellectual stimulation of transformational leadership, whereas corrective learning, which focuses on enhancement, productivity, and the improvement of existing competencies and models, is more compatible with transactional leadership. In relation to this, this first hypothesis is proposed:

**Hypothesis 1 (H1).** *Intellectual stimulation induced by a leader will be positively related to the intellectual engagement of their followers*.

Job performance is the degree to which a worker contributes to the effectiveness of the organization, according to the expectations associated with their job role [27]. Organizations often give descriptions of the tasks that each worker must perform in their workplace and the objectives they have to achieve [28]. However, through behaviors that are not directly related to the described functions of the job, such as overtime, employees can expand and improve the organization. According to Goodman and Svyantek [28] (p. 254) “these behaviours are important because they form the organizational, social and psychological context for the activities and procedures of the task”.

When the level of intellectual engagement in employees is increased, the degree of participation is enhanced, which leads to better performance and a broader use of the intellect to improve work-related skills [29]. In research carried out to date, intellectual engagement has shown the strongest relationship with academic performance in students [30], but few papers have addressed the influence of this dimension of engagement on employee performance. In a recent study, Kharbat and AlSoud [31] suggested that in the work context, the relationship between intellectual engagement and performance can be positive, since the former increases the employee’s skills by stimulating their intellect. In relation to this, the second hypothesis is proposed:

**Hypothesis 2 (H2).** *Intellectual engagement is positively related to performance*.

However, the relationships between the variables that make up the organizational context are not direct. The JD-R theory proposes that the relationships between the dimensional variables of transformational leadership, engagement and performance are affected by each other and by other factors existing in the organizational context. In this context, Salanova et al. [32] carried out a study with hospital workers, and found that the influence of transformational leadership on performance was completely mediated by work engagement. In addition, in various scientific studies it has been hypothesized that transformational leadership has a positive indirect effect on performance, proposing work engagement as one of the possible mediators [32,33,34]. In line with these results, the third hypothesis is proposed:

**Hypothesis 3 (H3).** *Intellectual engagement positively mediates the relationship between intellectual stimulation and worker performance*.

To complete the research model of the current study, the JD-R theory is used to visualize how motivation is not only influenced by positive elements of the work environment. Authors such as Fernet et al. [35] found that transformational leadership could also be negatively affected by job demands. However, when conceptualizing the role of job demands, recent research has made a distinction between job demands as a hindrance and job demands as a challenge [36]. Job demands, when conceived as obstacles, are defined as work circumstances that involve excessive or undesirable work conditions and interfere with or inhibit an individual’s capability to achieve their objectives [37]. On the other hand, job demands perceived as a challenge are defined as aspects that require effort, but that potentially promote employees’ personal growth and their perception of effectiveness [5]. When employees experience high job demands, their perception of leader’s support, inspiration and quality training can be reduced [12,38], affecting the way in which the leader’s behaviors interact with the emotional state and performance of employees.

The lack of information, or the presence of incompatible tasks, in the workplace can generate negative evaluations of employees and prevent them from fulfilling their duties, thus producing role conflict. This occurs when employees experience two or more sets of incompatible demands, or different expectations associated with a given role, in which the compliance with any of these impedes the fulfillment of the other(s) [39,40,41]. In line with JD-R theory [4], role conflict might inhibit employee intellectual engagement—e.g., the generation of new and useful ways of solving problems [42,43]. More precisely, role conflict may increase cognitive strain. In doing so, it limits the employees’ cognitive resources that are available for task engagement [44], and reduces the workers’ capacity to utilize their intellectual potential and skills by preventing them from focusing on the problem and finding new ways to solve them [45].

However, some authors have suggested that workers facing conflicting pressures, despite being affected by the potentially detrimental effects of role conflict, may react in a constructive way, namely, by approaching problems and tasks via intellectual engagement [46]. This research process has suggested that being involved in multiple roles, as in the case of role conflict, exposes employees to divergent viewpoints and to a wide range of information [47]. As a result, employees might have the opportunity to create novel solutions that would allow them to successfully manage complicated problems [48,49]. In this respect, the insights offered by activation theory [50] suggest that the level of role conflict faced by employees might be central to moderating their level of activation in response to work stressors (e.g., role conflict). Therefore, it might promote task engagement in the form of intellectual engagement, which can increase performance and ensure the optimal use of cognitive resources [41,51].

It is advisable to study whether stress factors such as role conflict, accompanied by other good business practices, can motivate employees. This improvement could help to temporarily prevent a deficit in the achievement of objectives that the organization failed to complete on time or in which they could not clearly establish the role of each worker, generating certain levels of stress for employees even when they feel supported by their superiors. When employees experience moderate role conflict, the perception of support, inspiration and quality training given by the leader can improve their intellectual engagement by making them perceive their work as a challenge and involving them more in the tasks and objectives of the organization [12,52]. In this sense, transformational leadership behaviors, such as intellectual stimulation, can improves the performance of the workers, allowing them to develop their full potential [53,54]. These leadership behaviors motivate workers to achieve better results [55] and work beyond what is expected of them, providing a favorable work environment and creating a motivational process that leads to a context of moderate role conflict [52,56].

**Hypothesis 4 (H4).** *Role conflict positively moderates the effect of intellectual engagement arising from the interaction between leader intellectual stimulation and performance*.

In general, the current research focuses on whether labor demands such as role conflict may be necessary to ensuring that the leader’s behaviour positively influences the motivational process by improving work engagement [57], but research on how job demands moderate the relationship between leadership behaviors and work engagement is scarce. Transformational leaders inspire employees by fostering new approaches to different problems, improving social cohesion among employees, and addressing each person’s developmental needs and concerns by providing psychosocial support [13]. However, if the work environment does not pose challenges for workers, the effects of the leadership can be limited.

All of our study’s objectives, which were established with reference to job demands–resources theory, aimed to analyze the influence of intellectual stimulation on performance via its effect on intellectual engagement, as well as the necessity of role conflict. Based on a review of the literature, a moderated mediation model was proposed and tested (see Figure 1). It was postulated that intellectual engagement mediates the relationship between intellectual stimulation and performance. In addition, the effect of this mediation will be moderated by the level of role conflict, increasing the mediating effect of intellectual engagement on the influence of the leader’s intellectual stimulation on job performance. With this work we offer detailed information on the role that the leader should play within the organization in specific stressful situations, that is, when there is role conflict. Therefore, with the results of this work we could determine whether the role of the leader is fundamental in reducing employee stress.

## 2. Materials and Methods

### 2.1. Sample

The sample in this study comprised 705 employees of a private multinational company (response rate: 82.3%). This company produces and distributes innovative surfaces of high value in the fields of design and architecture around the world, and it is based in the province of Almeria (Spain). Men constituted 91.1% of the sample, while women comprised 8.8%. In terms of age, 22.5% were between 18 and 25 years old, 31.2% were between 26 and 35 years old, 29.8% were between 36 and 45 years old, 15% were between 46 and 55 years old and 1.4% were more than 56 years old. The most represented academic level was secondary education, at 59.3%, followed by higher education, at 22.7%, while 8.5% of the participants had university education, 2.4% had a Master’s degree, 1% had a PhD degree and 6.8% had other studies. In terms of seniority in the company, 64.4% of the participants had less than 5 years’ experience, 12.2% had between 5 and 10 years, 13.9% had between 10 and 15 years and 9.2% had over 15 years. The most represented type of contracts was full-time undefined for 62.5% of the participants, while 28% had a temporary full-time contract, 4.4% had a temporary part-time contract, 3% had a part-time indefinite contract and 2% had other types of contract.

The only selection criteria for participants was that they should have a job in the company in which the questionnaire was applied at the time of the study.

### 2.2. Measures

Leadership intellectual stimulation was assessed via three items of the intellectual stimulation sub-dimension of the transformational leadership scale [17], validated in Spain by Salanova et al. [58]. The respondents answered using a 7-point Likert-type scale ranging from 0 (totally disagree) to 6 (totally agree). An example of an item is “Our supervisor... has ideas that have forced us to rethink some things that we have never questioned before”. The Cronbach’s Alpha coefficient of this scale was 0.91.

Role conflict was measured using a three-item scale (e.g., “I receive incompatible requests from two or more groups of people”) derived from Rizzo et al. [6], translated into Spanish by Peiró et al. [59]. For all items, the answers represented a Likert-type format of 5 points, ranging from 1 (strongly disagree) to 5 (strongly agree). An example item is “I receive orders without having the human resources to carry them out”. The Cronbach’s Alpha coefficient for this scale was 0.86.

The Spanish version of the Intellectual, Social, Affective Engagement Scale (ISA Engagement Scale) [60] was used based on the development and application of the work engagement scale of Soane et al. [22]. The intellectual engagement scale is composed of 3 items. For all items, the answers represented a Likert-type format of 7 points, ranging from 1 (totally disagree) to 7 (totally agree). An example item is: “I focus hard on my work”. The Cronbach’s Alpha coefficient of this scale was 0.86.

For performance, we used the scale of Goodman and Svyantek [28]. The subjects were asked to think about their areas of work. This questionnaire is made up of 6 items. For all items the answers represented a Likert-type format of 7 points, ranging from 1 (totally disagree) to 5 (totally agree). An example item is “We achieved the objectives of the work”. The Cronbach’s Alpha coefficient of this scale was 0.83.

### 2.3. Procedure

The university ethics committees of the corresponding authors approved the study. The research team contacted the management of the company and explained the purpose of the project. Once they agreed to participate, the workers of each department were informed by their managers about the objective of the study and its relevance to the organization, emphasizing that their participation was crucial. The questionnaires were passed to all company employees in group sessions. The questionnaire was applied during working hours, in order to facilitate the management of both the groups and the work of the company. First, all the subjects were informed of the objectives and the development of the study verbally, to be able to give their consent. To answer the questionnaire, brief instructions were given before beginning. All doubts were addressed individually by the members of the research group present in the room. Confidentiality and anonymity in the processing of information was guaranteed through the use of codes in the questionnaires.

Mediation and moderation analyses were conducted with the non-parametric bootstrapping procedure to estimate direct and indirect influence using the PROCESS package in SPSS 26, as constructed by Hayes [61]. Hayes’ suggestion was followed and a multi-step mediation analysis was carried out. The influence of intellectual stimulation on performance (Model 4 in PROCESS) in the context of intellectual engagement and role conflict was analyzed, representing the mediator and the moderator, respectively (Model 7 in process). The indirect and conditional influence were deemed significant if the 95% bias-corrected (BC) bootstrap confidence intervals (CI) for those variables of influence, assessed based on 10,000 bootstrapped samples, were not through zero. Lastly, the Johnson–Neyman technique was used to derive the value of the moderator (role conflict) at which the influence of the predictor variable (intellectual stimulation) transitions from statistically significant to non-significant at an alpha level of 0.05.

## 3. Results

The means, standard deviations and correlations between the scales are presented in Table 1. The results of the correlation analysis show that intellectual stimulation was positively related to intellectual engagement and performance, and negatively to role conflict.

Table 2 shows the results of the models tested in the mediation analysis. In the first model, intellectual stimulation was a significant predictor of the mediator (intellectual engagement). According to the second model, the total influence of intellectual stimulation over performance was significant (Co = 0.074, SE = 0.023, *p* < 0.01). The indirect effect shows that 0.064 of the total influence is explained by the mediating effect of intellectual engagement.

Table 3 shows that moderation analysis identified that the effect of intellectual stimulation on the frequency of employee intellectual engagement was a function of the level of role conflict (interaction coefficient: intellectual stimulation × role conflict). Specifically, as is also shown in Table 3, analysis via the Johnson–Neyman technique indicated that the conditional influence of intellectual stimulation on the frequency of employees’ intellectual engagement is positive and significant, as suggested from the role conflict score of 1.82.

Finally, the research model examining the modulated mediational effect of role conflict in the relationship between intellectual stimulation, intellectual engagement and performance was assessed (Figure 2).

## 4. Discussion

The objective of this manuscript is to outline the necessity of job demands, such as role conflict, in ensuring the positive influence of policies in stimulating employees via leader engagement and performance. In other words, employees will feel intellectually engaged when their team leader induces intellectual stimulation and they perceive role conflict within the organizational environment. All this will positively affect the performance of employees.

In view of the results obtained, we can conclude that our hypotheses are supported. The first hypothesis (intellectual stimulation as induced by the leader is positively related to the intellectual engagement of followers) is confirmed, and it is further shown, in line with other studies [24,25,26], that intellectual stimulation is more compatible with creativity, taking greater risks and developing new ideas. Intellectual stimulation, which encourages creativity and problem-solving, would be reflected in intellectual engagement improvement.

The second hypothesis, that intellectual engagement is positively related to performance, is supported, as workers with high levels of intellectual engagement showed better performance levels. This may be due to the fact that these more engaged workers participated more, which leads to better performance, as the former increases the employee’s skills by stimulating their intellect [29,31].

With regard to the third hypothesis, that intellectual engagement positively mediates the relationship between intellectual stimulation and worker performance, our results adhere to those of other studies [32,33,34], stating that productive leadership as a work resource motivates workers by improving their intellectual engagement and thus their performance.

Finally, the fourth hypothesis of this study, that role conflict positively moderates the mediation effect of intellectual engagement between leader intellectual stimulation and performance, is supported. The literature suggests that workers facing conflicting pressures may react in a constructive way, inducing intellectual engagement [46]. The current results suggest that role conflict exposes employees to divergent viewpoints, and might produce the opportunity to create novel solutions to complicated problems [41].

Given the results we obtained, this study makes three main theoretical contributions. First, it confirms the mediating role of intellectual engagement in the relationship between intellectual stimulation by the leader and performance. Previous studies, such as those by Salanova et al. [32], Kovjanic et al. [33] and Zhu et al. [34], have demonstrated the relationship between leadership and performance, with work commitment as a possible mediator, but none have addressed this relationship focusing on the contribution of employee knowledge. Secondly, the study shows the necessity of challenging demands in the work environment in order to connect employees with the organization. The traditional study approach concerns how positive organizational practices, such as leadership, can reduce stress [12,41,51]. The present study also shows that role conflict, as a stressor, increases the influence of intellectual stimulation over intellectual engagement. This makes role conflict necessary to activate the influence of the leader’s behavior over the employee’s emotional response, which is supported by the theory regarding active jobs that generate engagement [9,10]. With regard to the need for active jobs, in any job or organization it will be necessary to maintain a certain controlled tension in employees. As a third theoretical implication, our research (among others) affirms that the level of conflict determines its role as a challenge or a demand [50]; according to this line of research, the conflict is only challenging at low and medium levels, and not at high levels. The results of the present work, as derived from the application of the Johnson–Neyman technique, show how high levels of conflict can also have a positive effect on engagement if they are supported by appropriate leadership practices.

This paper presents the following limitations. Firstly, there are limitations in the method that has been used, as the information was collected through a commonly used method [62] without any other type of information collection technique, such as interviews or the collection of objective data from the organization. Secondly, in future studies the sample could be extended to workers from other mixed gender samples, and other private or public sectors. Third, this study is cross-sectional, and provides less information than other types of studies, such as longitudinal studies, which allow analysis at greater depth and show the evolution and causality of the variables studied.

In future research, it would be interesting to use complementary tools, such as employing specific questionnaires with the leaders of the organization, personal interviews with employees, observations of the organization and collections of objective data from the organization (such as absenteeism of workers, productivity, etc.), as these approaches would provide more information and allow a greater depth of analysis. It would also be advisable to study the relationships between the variables of this study in other work environments in which workers face other negative and harmful factors, such as boredom or burnout, and relate them to other variables, such as work environment, job satisfaction, or psychological capital, in order to expand the study model.

## 5. Conclusions

This paper shows that when workers face a challenging or demanding situation, and they are stimulated by their leader, they may be pushed to come up with new ways of solving the problem, and thus improve their performance. With the increased use of technology, many jobs have become monotonous and conflicting, and it is necessary to address the daily life of workers to reduce the challenges they face. However, if companies provide employees with skills and stimulation by posing challenges, it may make them feel more useful and attend more to their work, improving their work engagement and their performance.

## Figures and Tables

**Figure 1 ijerph-18-03630-f001:**
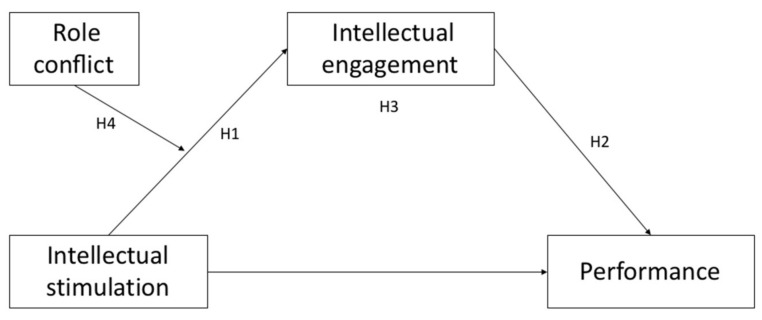
Research model; Own elaboration.

**Figure 2 ijerph-18-03630-f002:**
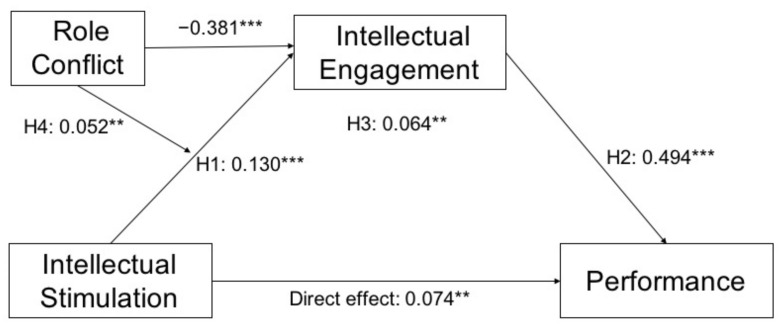
Data research model. Note: *** *p* > 0.001; ** *p* > 0.01.

**Table 1 ijerph-18-03630-t001:** Descriptive and correlations.

Variable	M	SD	1	2	3	4
Intellectual stimulation	4.72	1.12		−0.298 ***	0.448 ***	0.336 ***
Role conflict	2.49	0.72			−0.381 ***	−0.25 2***
Intellectual engagement	5.98	0.82				0.470 ***
Performance	6.03	0.81				

Note: *** *p* < 0.001.

**Table 2 ijerph-18-03630-t002:** Results from the regression analyses examining the mediator model of the influence of intellectual stimulation (X) on performance (Y) through intellectual engagement (M1).

Variables	Coefficient	SE	*p*
Model 1 (Intellectual Engagement)			
X (Intellectual Stimulation)	0.130	0.020	<0.001
Constant	5.806	0.098	<0.001
R^2^ = 0.056, F = 41.847, *p* ≤ 0.001			
Model 2 (Performance)			
X (Intellectual Stimulation)	0.074	0.023	<0.01
M (Intellectual Engagement)	0.494	0.041	<0.001
Constant	2.499	0.026	<0.001
R^2^ = 0.058, F = 43.453, *p* ≤ 0.001			
Indirect Effect (Performance)			
X (Intellectual Stimulation)	0.064	0.013	<0.01

Note: SE (Standard Error).

**Table 3 ijerph-18-03630-t003:** Results of regression analysis examining the moderation of the influence of intellectual stimulation on intellectual engagement by role conflict and the conditional influence of role conflict.

Antecedent	Coefficient	SE	*p*
X (Intellectual Stimulation)	−0.035	0.063	0.570
W (Role Conflict)	−0.381	0.100	<0.001
X × W	0.052	0.020	<0.01
Constant	6.979	0.318	<0.001
R^2^ = 0.007, F = 9.912, *p* = 0.008			
Johnson–Neyman Technique			
**Role Conflict Scores**	**Coefficient**	**SE**	**T**
1.00	0.016	0.044	0.376 n.s.
1.20	0.027	0.041	0.665 n.s.
1.40	0.037	0.037	1.006 n.s.
1.60	0.048	0.034	1.411 n.s.
1.80	0.059	0.031	1.893 n.s.
1.82	0.060	0.030	1.963 *
2.00	0.069	0.028	2.466 *
2.20	0.080	0.025	3.137 **
2.40	0.090	0.023	3.899 ***
2.60	0.101	0.021	4.711 ***
2.80	0.111	0.020	5.488 ***
3.00	0.122	0.020	6.114 ***
3.20	0.132	0.020	6.499 ***
3.40	0.143	0.021	6.629 ***
3.60	0.153	0.023	6.562 ***
3.80	0.164	0.025	6.379 ***
4.00	0.174	0.028	6.143 ***
4.20	0.185	0.031	5.894 ***
4.40	0.196	0.034	5.655 ***
4.60	0.206	0.038	5.434 ***
4.80	0.217	0.041	5.233 ***
5.00	0.227	0.045	5.054 ***

Note: *** *p* > 0.001; ** *p* > 0.01; * *p* > 0.05; n.s.: not significant.

## Data Availability

The data presented in this study are available on request from the corresponding author. The data are not publicly available due to privacy and ethical restrictions.

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
