# Peer review of "Are Job Demands Necessary in the Influence of a Transformational Leader? The Moderating Effect of Role Conflict"

_ijerph, 2021, doi:10.3390/ijerph18073630_

Round 1
Reviewer 1 Report
Dear Authors
I carefully evaluated the present paper, finding it overall interesting, providing relevant results on the relationship between job demand, leadership and performance. The theme is interesting, but the manuscript needs some improvements. Some concerns have to be solved before re-evaluating the manuscript and possibly considering it for publication. The language, specifically the grammar, is still not enough. There is the need to correct several inaccuracies and to improve reading fluency. At least at the end of the review rounds a proof-reading is recommended.
Introduction:
The introduction must be improved. In particular, the theoretical background needs strong improvements. The study framework lacks of a clear motivation to assess the relationship between your variables of interest. Despite you have described them singularly, the link between them remains unclear. I suggest to perform an in-depth literature search, in order to provide a clear study motivation, also referring to relevant scientific papers about the link between your variables of interest. In other terms, you should justify the reason why you choose those variables, and the motivation to construct such original framework. Moreover, state clearly what is the scientific gap you want to bridge with your research and why it is so important to publish data on this theme.
Methods:
The methods section is overall enough. Instruments were clearly described, but the analysis strategy should be described more in details.
Figure 1. I suggest to add the Hypothesis in the lines of the link between the variables (H1, H2, H3…)
Results:
the results section should be improved. I suggest to add a Table on the results of the moderate mediation model tested.
Discussion
The generalizability of your results should be improved. Are these finding relevant also out of a local context?
In the conclusion section, state clearly what this paper adds to knowledge about the theme, with respect to previous published articles. The novelty of your findings remains questionable. How your results contribute to bridge a literature gap?
Best regards
Author Response
Dear reviewer,
We are deeply grateful for his comments. They have undoubtedly been of great help in improving the manuscript. Next, I will answer each of the questions you have raised and present the changes made to the text. All these changes are highlighted in yellow in the article for easy review.
- To your first question: "The introduction should be improved. In particular, the theoretical background needs strong improvements. The study framework lacks a clear motivation to evaluate the relationship between its variables of interest. Although it has described them in a unique way, the link between them remains unclear. I suggest conducting an in-depth bibliographic search, in order to provide a clear motivation for the study, also referring to relevant scientific articles on the link between your variables of interest. In other words, you must justify the reason you choose those variables and the motivation to build that original framework. Also, clearly state what scientific gap you want to bridge with your research and why it is so important to publish data on this topic"
R: More current references and closer to the objective of the study have been introduced in the text. Paragraphs have been modified to better guide the reader around the link between the study variables and it has been tried to convey the objective of the work with greater clarity. All this with special emphasis on the first part of the introduction, which may be the one that will generate the greatest confusion.
- To answer your questions about the method:
Q1. "The methods section is generally sufficient. The instruments were clearly described, but the analysis strategy should be described in more detail."
R1: A new paragraph has been included that describes the analyzes in greater detail.
Q2: "Figure 1. I suggest adding the Hypothesis in the lines of the link between the variables (H1, H2, H3…)"
R2: The description of the hypotheses (H1, H2, H3 ...) has been added to the manuscript image.
- To answer your questions about the result: “the results section should be improved. I suggest to add a Table on the results of the moderate mediation model tested”
R: We have included the requested tables with the results and we have restructured the section to improve its understanding
In the last of your question,
Q1 “The generalizability of your results should be improved. Are these findings relevant outside of a local context as well?”
R1: Although more studies would be necessary in other organizational contexts, our hypothesis is that the result is generalizable from the theory of active jobs. Thus, in any job it is necessary to maintain a certain tension and pressure, as long as the employee has the resources and the necessary support to face them. It is also reflected in the new paragraph.
Q2. In the conclusions section, clearly indicate what this article contributes to knowledge on the subject, with respect to previously published articles. The novelty of their findings remains questionable. How do your results contribute to closing a gap in the literature?
R2: A more complete paragraph on the implications of this study has been added to the theory, indicating mainly 3: On the leadership-commitment-performance relationship, on the perception of conflict as necessary in the effects of the leader's behaviors and on the level of necessary conflict.
Thank you very much for the effort you have made in reviewing it, it has undoubtedly been of great use in improving the manuscript.

Reviewer 2 Report
The article presents the results of an empircial study rooted in the Job Demads-Resources theory. The research was conducted in Spain on a sample of 705 employees working in a production company. The authors tested the moderate mediation model to examine the moderating effect of role conflict for team leader’s intellectual stimulation on the employees’ intellectual engagement and performance. The aim of the paper is well defined and clearly explained. The hypotheses are properly anchored in scientific theory and competently justified. The study design is correct and the analyses are performed according to the best methodological standards.
The methods and tools are sufficiently described, although I suggest minor corrections in section 2.2. (e.g.: in line 219 the authors state: “ (…) was assessed by three items (…) while in line 222 they give an example of “Item 6 (…)” – it sounds incoherent. Thus, I suggest omitting the numeration of the cited items, as it does not matter and causes unnecessary confusion - this remark is valid for the all the example items cited in section 2.2. I also suggest that the manuscript is proof-read by a linguist (preferably a native speaker of English) as it seem to have some grammatical and punctuation errors.
Generally, the text is important and valuable both for scholars and practitioners. The conclusions are justified, coherent and convincing. Therefore, I recommend the text for publication, after minor amendments and linguistic revision.
Author Response
Dear reviewer,
We are deeply grateful for his comments. They have undoubtedly been of great help in improving the manuscript. Next, I will answer each of the questions you have raised and present the changes made to the text. All these changes are highlighted in yellow in the article for easy review.
- To your question:
The methods and tools are sufficiently described, although I suggest minor corrections in section 2.2. (e.g.: in line 219 the authors state: “ (…) was assessed by three items (…) while in line 222 they give an example of “Item 6 (…)” – it sounds incoherent. Thus, I suggest omitting the numeration of the cited items, as it does not matter and causes unnecessary confusion - this remark is valid for the all the example items cited in section 2.2. I also suggest that the manuscript is proof-read by a linguist (preferably a native speaker of English) as it seem to have some grammatical and punctuation errors.
R: Following your suggestion we have omitted the numbering of the cited items. Review of the article by a linguist has also been requested.
Thank you very much for the effort you have made in reviewing it, it has undoubtedly been of great use in improving the manuscript.

Round 2
Reviewer 1 Report
Dear Authors
All my concerns have been properly addressed.
Best Regards
Author Response
Dear reviewer,
Thanks for your review, it has certainly been useful to improve the manuscript
Kind regard